# Updates on Disease Mechanisms and Therapeutics for Amyotrophic Lateral Sclerosis

**DOI:** 10.3390/cells13110888

**Published:** 2024-05-21

**Authors:** Lien Nguyen

**Affiliations:** 1Department of Molecular Genetics and Microbiology, College of Medicine, University of Florida, Gainesville, FL 32610, USA; lien.nguyen@ufl.edu; 2Center for NeuroGenetics, College of Medicine, University of Florida, Gainesville, FL 32610, USA; 3Department of Neuroscience, College of Medicine, University of Florida, Gainesville, FL 32610, USA; 4Genetics Institute, University of Florida, Gainesville, FL 32610, USA

**Keywords:** SOD1 ALS, C9 ALS, TDP43, disease mechanisms, genetics, ALS FDA approved drugs, ALS treatment strategies

## Abstract

Amyotrophic lateral sclerosis (ALS), or Lou Gehrig’s disease, is a motor neuron disease. In ALS, upper and lower motor neurons in the brain and spinal cord progressively degenerate during the course of the disease, leading to the loss of the voluntary movement of the arms and legs. Since its first description in 1869 by a French neurologist Jean-Martin Charcot, the scientific discoveries on ALS have increased our understanding of ALS genetics, pathology and mechanisms and provided novel therapeutic strategies. The goal of this review article is to provide a comprehensive summary of the recent findings on ALS mechanisms and related therapeutic strategies to the scientific audience. Several highlighted ALS research topics discussed in this article include the 2023 FDA approved drug for *SOD1* ALS, the updated *C9orf72* GGGGCC repeat-expansion-related mechanisms and therapeutic targets, TDP-43-mediated cryptic splicing and disease markers and diagnostic and therapeutic options offered by these recent discoveries.

## 1. Introduction to ALS

Amyotrophic lateral sclerosis (ALS), also called Lou Gehrig’s disease, is characterized by the degeneration of upper motor neurons in the motor cortex and lower motor neurons in the spinal cord. The disease prevalence for ALS is 4–8 per 100,000, and is different among populations [1]. Males are reported to be more susceptible to ALS than females, with the prevalence of ALS being approximately 1.5 times higher in males than females [2,3,4]. The typical onset of ALS in patients is between 55 and 60 years of ages, with the incidence increasing with age.

ALS is a heterogeneous neurodegenerative disorder with a wide range of severity levels of clinical symptoms [5,6]. Some ALS patients develop focal muscle weakness and wasting of the upper or lower limbs, while others show difficulty with speech (dysarthria) and swallowing (dysphagia) at the initial stage of the disease, known as spinal or limb onset and bulbar onset, respectively. Approximately 60% of ALS patients present with spinal or limb onset [6,7]. A large variation in disease symptoms has been observed even within each disease onset group. The limb symptoms can start either distally or proximally in the upper and lower limbs and could appear simultaneously with bulbar onset.

There is growing evidence that ALS is a multisystemic disease with a major effect on the motor system. Approximately 50% of ALS patients develop frontotemporal dementia (FTD)-related features that are neuropathologically characterized by neuronal loss in the frontal and temporal lobes and changes in behavior and judgement abilities [8,9]. Additionally, autonomic disturbances and peripheral sensory abnormalities were reported in a substantial group of ALS patients. These non-motor symptoms progress in parallel with motor symptoms and can be prognostic biomarkers of ALS [10,11]. The reported mean survival time for ALS patients from the clinical diagnosis is 3–5 years, although disease outcomes largely vary between patients. Approximately 10–20% of ALS patients have more than 10 years survival after the first clinical symptom appears. The most frequent cause of death in ALS patients is respiratory failure due to muscle paralysis that develops progressively, with a mean of 2–3 years for bulbar-onset cases and 3–5 years for spinal-onset cases [12].

Approximately 10% of ALS cases present with inherited familial forms of the disease (fALS), while the remaining cases develop a sporadic form of the disease (sALS) with unknown family histories of the disease. A number of genetic mutations in *SOD1*, *C9orf72*, *TARDBP* and *FUS* and very rare mutations in more than 40 other genes have been linked with ALS. These mutations are estimated to explain approximately 60% and 40% of fALS cases in European and Asian populations, respectively. Among these mutations, the *C9orf72* GGGGCC (G4C2) repeat expansion is the most common known genetic cause of ALS, explaining approximately 40–50% of fALS in Europe and the USA’s populations [13]. Compared to fALS, the genetics of sALS are far less understood. The *C9orf72* G4C2 repeat expansion is found in approximately 5–10% of sALS cases [14], while SOD1 mutations have been shown to be present in approximately 2% of sALS cases [15]. These findings, combined with the predicted heritability of ALS reported in a twin study, population-based study, and register study of 40–60%, strongly suggest that other genetic factors of ALS remain to be identified [16,17,18]. In addition to genetic factors, lifestyle choices and environmental exposures were shown to have links with ALS, and were discussed in detail in other review articles [19,20]. Neuropathologically, ALS is characterized by the loss of motor neurons, the accumulation of TAR DNA-binding protein TDP-43 and p62 inclusions. The details of these ALS pathological hallmarks, including their distribution and correlation with the disease, were discussed elsewhere [21].

Treatments available for ALS include Nuedexta, Exservan, Tiglutik, Rilutek, Radicava, RELYVRIO, and QALSODY^®^, with most of them acting to slow down the disease progression of ALS patients [22] (Table 1). Four out of six ALS drugs were approved by the FDA in the last 6 years (2017–2023). Rilutek, the oral tablet form of Riluzole was the first FDA-approved drug for ALS. Riluzole is a benzothiazole molecule that inhibits the release of glutamate. Elevated glutamate levels in cerebrospinal fluid (CSF) were detected in approximately 40% of ALS cases and correlated with more severe spinal symptoms [23]. The thickened liquid and oral film forms of Riluzole, Tiglutik and Exservan were later developed and approved by the FDA in 2018 and 2019, respectively. Nuedexta, the first FDA-approved drug for pseudobulbar affect, is a combination of quinidine sulfate and dextromethorphan hydrobromide that regulates neurotransmission in the central nervous system and showed positive effects on bulbar functions in ALS patients [24,25]. Radicava, a free-radical scavenger, acts to remove reactive oxygen species and reduce oxidative stress that was shown to be involved in ALS pathogenesis [26]. RELYVRIO, an oral combination of sodium phenylbutyrate and tauroursodeoxycholic acid that also targets reactive oxygen species and oxidative stress, is one of the latest FDA-approved drugs for ALS. However, the recent phase III clinical trial of RELYVRIO did not support its clinical benefits (https://alsnewstoday.com/news/als-treatment-relyvrio-fails-trial-may-be-withdrawn-market/, accessed on 5 May 2024). The most recent FDA-approved drug for *SOD1* ALS (QALSODY^®^) was discussed in detail in the next section.

This article discusses five major topics in ALS: *SOD1* ALS, *FUS* ALS, *ATXN2* in ALS, *C9orf72* ALS and TDP-43 in ALS. The recent findings in these topics in the past decade and how they led to the identification of promising therapeutic targets and drug candidates for various forms of ALS including *C9orf72* ALS and TDP-43, were discussed in this article.

## 2. *SOD1* ALS

**Superoxide dismutase 1.** The *superoxide dismutase 1* (*SOD1*) gene, or *superoxide dismutase [Cu–Zn]* gene, encodes for the superoxide dismutase 1 protein that binds to copper and zinc molecules to catalyze the conversion of reactive oxygen species to hydrogen peroxide and oxygen [27]. Other functions beside the canonical antioxidant activity of SOD1 were also previously described [28], which includes acting as a sensor and switch to activate the endoplasmic reticulum (ER) stress response under Zn-deficient conditions [29], a nuclear transcription factor that regulates the expression of genes responsible for oxidative stress resistance and repair [30] and a key component of caspase-1 activation that regulates the expression of caspase-1-dependent cytokines [31].

***SOD1* mutations in ALS patients.** The first mutations in *SOD1* that cause ALS were identified in 1993 [27,32], making *SOD1* the first gene to be linked with ALS. Up till now, more than 180 ALS-associated *SOD1* variants were identified and these mutations were found throughout five exons of the *SOD1* gene [33]. The majority of ALS linked *SOD1* mutations are missense mutations that lead to alternation in amino acids in SOD1 protein. Additionally, nonsense mutations, frameshift mutations and deep intronic splicing mutations in *SOD1* can produce mRNA containing premature termination codons that result in truncated SOD1 proteins if escaping nonsense-mediated mRNA decay [34]. A wide range of clinical symptoms were observed in ALS cases carrying *SOD1* mutations, from the classic ALS phenotypes with the involvement of upper and lower motor neurons to progressive muscular atrophy to mixed phenotypes of ALS and neurodegenerative diseases [15]. Specific SOD1 mutations were found to be associated with distinct clinical phenotypes [35,36,37,38]. For example, the G41S mutation showed a significantly shorter survival time and faster progression while the L144S SOD1 mutation showed a longer survival and slower disease progression compared to other tested mutations, such as K3E, L126* and N139D [15]. Although there is a large number of ALS-associated *SOD1* mutations, it is not clear whether they produce a compounding contribution to the disease.

**Disease mechanisms of *SOD1* ALS.** The pathological hallmark of SOD1 ALS is the aggregation of misfolded SOD1 proteins that can transmit between cells. The propagation mechanisms of misfolded SOD1 proteins are still under investigation, but could involve exosome-dependent or independent mechanisms [39]. Most ALS patients carrying *SOD1* mutations lack the TDP-43 pathology that presents in more than 95% of ALS cases [40]. This observation suggests that the pathogenesis leading to the loss of motor neurons in *SOD1* ALS patients is different from other groups of ALS patients. *SOD1* mutations were shown to affect the dismutase enzymatic activity of SOD1, thus, the first proposed mechanism for *SOD1* ALS was the loss of function. However, several studies showed that the loss of the SOD1 protein was not required with motor phenotypes observed in *SOD1* ALS mouse models. The first transgenic mouse model of an ALS-associated *SOD1* mutant, SOD1-G93A mice that carry a high copy number of mutant *SOD1*, were shown to develop progressive motor phenotypes starting at 90 days of age with minimal effects on the enzymatic activities of SOD1 protein [41]. The severity of behavioral deficits and neuropathology of SOD1-G93A mice were associated with the copy numbers of SOD1 mutants [42]. The SOD1-G93A mice developed SOD1 protein aggregates in motor neurons that recapitulated the pathological hallmark in most *SOD1* ALS patients. The mutations, including homozygous D92G-*SOD1*, showed less aggregates and a lower toxicity compared to G93A-*SOD1* or L1144FVX-*SOD1*, and were linked with less severe disease in patients [35]. In contrast, *SOD1* knockout mice developed normally with no overt motor phenotypes by 6 months of age and no sign of disease pathology by 4 months of age. The *SOD1*^+/−^ and *SOD1*^−/−^ mice showed an increased loss of motor neurons after axotomy, suggesting the involvement of SOD1 in axonal injury and neuronal dysfunction [43]. Thus, these results support a gain of function mechanism of *SOD1* ALS mutations.

The discovery of novel ALS *SOD1* mutations suggests the possible contribution of SOD1 loss of function to the progression and severity of ALS. Fahmy and co-workers reported a homozygous p.Ser69Pro *SOD1* mutation that caused a ~80% decrease in SOD1 enzyme activity and severe young-onset ALS [44]. Andersen et al. reported an infant with a severe motor neuron dysfunction who carried homozygous duplication c.335dupG *SOD1*. This mutation resulted in a truncated SOD1 protein with a complete lack of activity in the patient’s erythrocytes and a high sensitivity to oxygen in the patient’s cell culture [45]. The work by Çakar and co-worker identified a homozygous variant (c.52_56del5ins154) in exon 1 of *SOD1* that caused truncated SOD1 mRNA and a ~100% loss of SOD1 enzymatic activity in erythrocytes in a 22-month-old boy with a loss of motor functions [46]. Additional studies are needed to further understand the toxicity of these recently identified mutations and how they contribute to ALS pathogenesis.

**Antisense oligo molecules for *SOD1* ALS.** Given the gain of function mechanism of *SOD1* ALS mutations shown in cellular and mouse models, scientific efforts have focused on developing therapeutic molecules that lower SOD1 mRNA and SOD1 protein levels. Multiple antisense oligos containing 20 nucleotides were developed to target the coding regions in *SOD1* via a Watson–Crick base pairing. The resulting SOD1 RNA–DNA antisense heteroduplex was recognized and the target SOD1 mRNA was then degraded using endonuclease RNaseH [47]. An infusion of SOD1 antisense oligos reduced SOD1 mRNA and SOD1 protein levels to approximately 5–10% and 40% of the normal levels in rat livers and frontal cortex or spinal cord, respectively. The most promising first-generation antisense oligo SOD^r/h333611^ targeting the first exon of SOD1 slowed down disease progression in SOD1-G93A rats and increased the mean survival of approximately 10 days [48]. While a phase I study demonstrated a safety profile of this first class of SOD1-targeting ASOs [49], more potent molecules were needed to improve the efficacy. The second class of SOD1 antisense oligos was identified through an *in vitro* screen of more than 2000 ASOs targeting SOD1. An intracerebroventricular injection of the most potent second-generation ASO molecules targeting the 3′ untranslated region (UTR) of *SOD1* resulted in a greater reduction in SOD1 mRNA and protein levels compared to the first-generation SOD1 ASO [50]. The treatment of these second-generation SOD1 ASOs significantly increased the survival of SOD1-G93A rats (more than 50 days) and mice (40 days), reversed muscle phenotypes and reduced the phospho-neurofilament heavy chain (pNFH) serum levels in SOD1-G93A mice. One of the second-generation SOD1 ASOs, Tofersen or BIIB067, was further studied in clinical trials supported by Biogen. A phase 1-2 clinical trial showed that a 12-week intrathecal administration of three different doses of Tofersen resulted in an approximated 33% reduction in SOD1 protein levels in cerebrospinal fluid (CSF) compared to the baseline at the highest dose (100 mg) [51]. The phase 3 clinical VALOR trial of Tofersen showed a greater reduction in CSF SOD1 protein and plasma neurofilament light-chain (Nfl) levels, a general marker of neuronal damage [52,53] at 28 weeks. However, there was no significant improvements in clinical outcomes in patients with faster-progressing patients measured using the total score on the ALS Functional Rating Scale–Revised (ALSFRS-R), although a trend of clinical improvements was suggested [54]. The open label extension of the VALOR trial showed a consistent reduction in plasma Nfl levels and continued to suggest some beneficial outcomes including respiratory function and survival time at 40 weeks (https://biogen.gcs-web.com/static-files/b2154d4e-f69f-49d4-9a61-e834387293ea, accessed on 5 May 2024 [55]). The Food and Drug Administration (FDA) approved Tofersen (QALSODY^®^) via an accelerated mechanism on 25 April 2023, making this drug the first gene therapy for ALS. A multicenter ALS cohort study of Tofersen in Germany showed consistent positive effects of this drug on lowering serum NfL and CSF pNfH levels in *SOD1* ALS patients. Tofersen was recommended for European approval in Feb 2024 to treat *SOD1* ALS (https://www.ema.europa.eu/en/medicines/human/EPAR/qalsody, accessed on 5 May 2024). This exciting achievement for the ALS community was followed by multiple questions to address: Does QALSODY^®^ show significant clinical improvements that can be measured with the ALSFRS-R or any other alternative scales? Or do we need a more potent and safer molecule for *SOD1* ALS? Is this treatment safe for a long-term treatment given some adverse effects and immune response detected in a subset of patients upon treatment [51,54,56]? Due to more severe and earlier onset of *SOD1* mutations that lead to the complete loss of SOD1 enzymatic activity being described, it would be important to monitor patients to avoid unwanted effects of a long-term treatment. Personalized treatment strategies, including the alternation of dose, treatment period and start time, may be needed for the optimal clinical benefits of QALSODY^®^ for *SOD1* ALS patients. Related to these topics, ATLAS, an ongoing randomized, placebo-controlled, phase 3 trial, aims to study if Tofersen can delay the manifestation of ALS symptoms in presymptomatic SOD1 variant carriers [57]. Additionally, other interventions including an adeno-associated virus and microRNA system, were developed to target *SOD1* ALS, which could be potential therapeutic strategies but require further development [58].

## 3. *FUS* ALS

**Fused in sarcoma.** *Fused in sarcoma* or the *FUS* gene, was first identified as a fusion oncogene in human liposarcomas [59,60]. *FUS* encodes for the FUS protein, a nuclear DNA–RNA-binding protein that is rich in serine, tyrosine, glycine and glutamine and belongs to the FET family. FUS is a multidomain protein containing a prion-like or low-complexity N-terminus domain, an RNA recognition motif, glycine-rich domains, a C2C2 zinc finger motif, and a nuclear localization signaling (NLS) C-terminal [61,62]. This protein was found predominantly in the nucleus and involved in regulating transcription, alternative splicing, mRNA transport, stability and translation [63,64,65,66,67]. The FUS protein was also reported to play roles in DNA damage repair [61,68,69,70]. The physiological functions of FUS proteins were discussed in detail in other review articles [71,72].

***FUS* mutations in ALS.** The first ALS-associated *FUS* mutations were reported in familial ALS cases in 2009 [73,74]. *FUS* mutations explain approximately 4–6% of fALS and less than 2% of sALS patients [74,75]. Currently, more than 50 missense and deletion/insertion *FUS* mutations are known to link with ALS, with the majority of them located in the prion-like domain, glycine-rich region or the NLS C-terminus [76]. Three major FUS ALS groups were reported by Grassano et al. corresponding to three genotype–phenotype clusters: axial, benign and juvenile FUS ALS [77]. Among these groups, the juvenile FUS ALS group showed the most aggressive form of disease, with the age of onset at childhood or early adulthood, rapid respiratory failure, intellectual disability, learning impairment and tremors in a subset of cases. A large group of mutations in the FUS NLS C-terminus was linked with this most severe form of FUS ALS [77].

**Mechanisms of *FUS* ALS.** FUS-positive cytoplasmic inclusions in neuron and glial cells were reported in the spinal cord of FUS ALS cases and ALS cases without *FUS* mutations [78]. The cytoplasmic miscolocalization of FUS protein was also detected in other forms of ALS and 5–10% of FTD cases, suggesting a common pathogenic pathway of FUS dysregulation in ALS [79,80]. Two major proposed mechanisms of *FUS* ALS are the loss of FUS function in the nucleus and gain of function in cytoplasmic FUS inclusions [81,82,83,84]. A transgenic mouse model expressing the FUS-R521C mutant protein developed neurological dysfunction and motor abnormalities, including hindlimb paralysis, muscle wasting, and decreased survival [85]. A knockin mouse model expressing mislocalized cytoplasmic FUS protein was reported to develop respiratory phenotypes and decrease lifespan and motor neuron survival [82]. In contrast, knockout FUS mice showed decreased survival but did not develop motor neuron degeneration [82,86]. Another knockin mouse model of the FUS-R521C ALS mutation showed a progressive and selective loss of motor neuron and neuromuscular junction abnormalities while postnatal FUS removal did not affect motor neuron survival or function [86]. Additionally, FUS knockin mice expressing FUSP525L and FUSΔEX14 mutant protein developed progressive, age-dependent motor neuron degeneration [84]. ALS-related phenotypes in these mouse series were dose-dependent on the toxicity and insolubility of mutant FUS proteins. These results demonstrated the important contribution of the gain of function of cytoplasmic mislocalized FUS protein in *FUS* ALS. Recent studies have provided insights into the molecular mechanisms of FUS mutations in motor neuron degeneration. Mutations in FUS were shown to alter the proteins important for regulating RNA metabolism in neurons, for example, the Fragile X mental retardation protein (FMRP) and HuD/ELAVL4 proteins [87,88]. These alternations led to an upregulated synaptic protein synthesis and synaptic dysfunction in FUS ALS patient derived neurons. In addition to autonomous motor neuron loss, other brain cells were reported to be involved in disease pathogenesis of FUS ALS [89,90,91]. Co-culturing astrocytes overexpressing the R521G FUS mutant protein or FUS ALS patient derived astrocytes with motor neurons led to decreased motor neuron survival [89,90]. FUS ALS mutations were reported to alter astrocyte homeostasis that led to the increased secretion of inflammatory cytokines and triggered an inflammatory response in co-cultured motor neurons.

**Therapeutic strategies for *FUS* ALS.** With the scientific results supporting the gain in the function mechanism of cytoplasmic FUS inclusions, reducing FUS protein levels is one of the focused-on therapeutic strategies for *FUS* ALS. The treatment of Jacifusen or ION363, an antisense oligonucleotide targeting FUS in non-allele-specific manners, significantly reduced FUS protein levels in the brain and spinal cord and delayed motor neuron degeneration in *FUS* ALS knockin mouse models [84]. The administration of ION363 in a juvenile-onset *FUS* ALS patient carrying the FUSP525L mutation via repeated intrathecal infusions with increased monthly doses from 20 mg to 120 mg over 10 months led to a substantial reduction in wild-type and mutant FUS expression throughout the central nervous system. The cytoplasmic FUS protein aggregation also decreased to nearly non-detectable levels in a patient’s lumbar spinal cord autopsy. These encouraging results led to an on-going global multicenter phase III clinical trial on ION363 and *FUS* ALS (https://clinicaltrials.gov/study/NCT04768972, accessed on 5 May 2024). Regulating FUS protein localization by targeting FUS post-translational modification or FUS nucleocytoplasmic transport machinery was also shown as a potential therapeutic strategy for *FUS* ALS and was discussed in detail elsewhere [92].

## 4. *ATXN2* and ALS

Ataxin-2 is a RNA binding protein important for regulating translation. The details on ataxin-2 biology and function were discussed in other review articles [93,94]. The link of ataxin-2 with ALS was identified in an unbiased yeast screen for modifiers of TDP-43 toxicity [95]. The upregulation of ataxin-2 led to the increased toxicity of TDP-43 in *Drosophila*. Cytoplasmic ataxin-2 inclusions were detected at higher levels in the spinal cord of ALS patients compared to control cases. The CAG repeat expansion in *ATXN2* with repeat lengths of ≥35 CAG repeats is a cause of spinocerebellar ataxia type 2 (SCA2) [96]. The *ATXN2* CAG repeats with intermediate lengths of 24–34 repeats were found to increase ALS risk, with a stronger association detected for higher repeat length cutoffs [95,97,98]. In addition to ALS, *ATXN2* CAG repeats with intermediate lengths were also link with other diseases, including FTD and Parkinson’s disease [99,100,101]. *ATXN2* intermediate length CAG repeats were thought to decrease ataxin-2 turnover, resulting in increased axatin-2 levels and favoring the formation of ataxin-2 aggregates [95]. Intermediate-length *ATXN2* CAG repeat expansions were reported to increase the accumulation of a phosphorylated TDP-43 C-terminal fragment, caspase activation upon stress and FUS-related pathology in cellular models of the disease [102,103].

**Targeting *ATXN2* for ALS treatment.** Due to the link between ataxin-2 levels and TDP-43 related toxicity, strategies to reduce ataxin-2 levels were developed to mitigate TDP-43 pathology in ALS. Crossing ataxin-2 knockout mice with TDP-43 transgenic mice significantly increased the survival of TDP-43 transgenic mice in an ataxin-2 dose-dependent manner and reduced TDP-43 accumulation in stress granules and TDP-43 pathology. The treatment of an ASO targeting ATXN2 increased the lifespan and improved the motor performance of TDP-43 transgenic mice [104]. These promising preclinical studies led to the clinical trial on the *ATXN2*-targeting ASO, BIIB105 to reduce ataxin-2 in ALS patients with or without intermediate length *ATXN2* CAG repeats (https://alsnewstoday.com/news/biib105-trial-enrolling-als-with-without-atxn2-mutations/, accessed on 5 May 2024). In addition to the ASO approach, a CRISPR/Cas13-based system was also developed to target ATXN2 mRNA to lower ataxin-2 protein levels. The administration of a CRISPR/Cas13-based system targeting ATXN2 in mice expressing TDP-43 pathology increased the survival, improved the functional deficits and decreased the neuropathological hallmarks of these mice [105].

## 5. *C9orf72* ALS/FTD

***C9orf72* GGGGCC repeat expansion and ALS/FTD**. The chromosome 9p21 was first identified as a risk locus that could link ALS and FTD in 2006 through the linkage studies on families with autosomal dominant ALS and FTD [106,107]. The identification of the hexanucleotide GGGGCC repeat expansion (G_4_C_2_^exp^) in the first intron of *C9orf72* as a genetic cause of ALS and FTD in 2011 was a breakthrough discovery that linked ALS and FTD to a larger group of repeat expansion disorders [108,109,110,111]. The threshold of the pathogenic repeat length of *C9orf72* G_4_C_2_^exp^ has not yet been clearly defined but a wide-range of expanded repeat lengths have been detected in C9 ALS/FTD patients [112]. The *C9orf72* G_4_C_2_^exp^ mutation was found to account for approximately 10–50% of fALS, 5–7% of sALS, 12–25% of fFTD, and 6–7% of sFTD depending on the population [113]. *C9orf72* G_4_C_2_^exp^ carriers without disease symptoms were also reported. The penetrance of *C9orf72* G_4_C_2_^exp^ is incomplete and increases with age. The penetrance of the pathogenic *C9orf72* G_4_C_2_^exp^ with repeat lengths of >30 repeats was reported to be 50% and almost 100% by the age of 58 and 80 years, respectively [13]. These results suggested the involvement of other factors such as aging, the environment and lifestyle in C9ALS/FTD [114,115]. The fact that *C9orf72* G_4_C_2_^exp^ is a common genetic cause for ALS and FTD (C9ALS/FTD) demonstrated a high heterogeneity of the clinical symptoms manifested by this mutation. Even within the family, disease onset, progression rate, and severity highly vary between carrier individuals. The mean survival of C9ALS is approximately 2.8 years compared to 4 years of total ALS patients [116]. Tremendous scientific efforts on understanding the pathogenesis of C9ALS/FTD led to important findings of disease molecular mechanisms that were discussed in details elsewhere [117,118]. This section highlighted several key proposed mechanisms and therapeutic intervention strategies for C9ALS/FTD.

**C9orf72 loss of function**. The first proposed mechanism of C9ALS/FTD is C9orf72 protein haploinsufficiency due to the G4C2 repeat expansion. A reduction in exon 1b containing a RNA variant and C9orf72 protein levels was detected in postmortem frontal cortex tissues from C9ALS/FTD patients [119]. At the time that *C9orf72* G_4_C_2_^exp^ was identified, the function of the C9orf72 protein was largely unexplored, and efforts on studying C9orf72 haploinsufficiency in C9ALS/FTD showed that C9orf72 plays roles in the immune response. Studies showed that C9orf72 regulates inflammation via JAK–STAT and is involved in vesicle trafficking and the lysosomal degradation of inflammatory mediators, including Toll-like receptors (TLRs) and cGAS-STING [120,121]. C9orf72-deficient mouse lines developed immune phenotypes characterized by myeloid expansion, T cell activation and increased plasma cells, or progressive splenomegaly and lymphadenopathy with an accumulation of engorged macrophage-like cells, but no motor phenotypes [122,123]. Only FTD-like phenotypes and mild motor phenotypes were observed in a C9orf72 knockdown model that was generated through a microRNA-based knockdown approach [124]. Recent studies have suggested that C9orf72 protein haploinsufficiency may interplay with other pathogenic pathways in C9ALS/FTD to exacerbate disease phenotypes [125,126,127,128]. In summary, while C9orf72 haploinsufficiency is not a cause of motor phenotypes in C9 ALS, it could play some roles in the manifestation of specific disease phenotypes and disease progression.

***C9orf72* repeat expansion RNA gain of function**. Like other repeat expansion mutations, the *C9orf72* G_4_C_2_^exp^ mutation can be transcribed in both directions [129]. The resulting sense rG_4_C_2_^exp^ can adopt hairpin and G-quadruplex structures while the antisense rC_4_G_2_^exp^ can form hairpin structures [130,131,132]. Both sense rG_4_C_2_^exp^ and antisense rC_4_G_2_^exp^ RNA transcripts form foci that were detected in patient postmortem brain tissue and patient cell lines [133,134,135]. However, C9 sense and antisense expansion RNAs showed different biological properties and accumulated in patient tissues with different patterns [136]. For example, antisense rC_4_G_2_^exp^, but not sense G_4_C_2_^exp^, RNA was found to be associated with TDP-43 pathology and the PKR/eIF2α-dependent integrated stress response (ISR) [136,137]. The rG_4_C_2_^exp^ and rC_4_G_2_^exp^ RNAs were shown to sequester multiple RNA-binding proteins to RNA foci and contribute to disease via the RNA gain-of-function mechanism [138].

**RAN proteins in C9 ALS/FTD**. The *C9orf72* G_4_C_2_^exp^ mutation can also cause disease via toxicity of mutant expansion proteins that can be expressed via repeat-associated non-AUG (RAN) translation [139]. Polymeric glycine–proline (GP), glycine–alanine (GA) and glycine–arginine (GR) can be expressed from sense rG_4_C_2_^exp^, and polymeric proline–arginine (PR), proline–alanine (PA) and glycine–proline (GP) can be produced from antisense rC_4_G_2_^exp^. C9 RAN polymeric dipeptide aggregates were detected with different accumulation levels and patterns in postmortem brain and spinal cord tissue samples from C9 ALS/FTD patients [134,140,141]. PolyGP, polyGA, and polyGR were also detected in CSF from C9 patients, and could be used as prognostic markers of the disease [142,143,144]. Among C9 dipeptide proteins, polyGA aggregates were found to be the most abundant in C9 autopsy brains, and polyGR aggregates were reported to correlate with neurodegeneration and clinicopathological subtypes of C9 patients [145]. Mechanisms of C9 RAN dipeptide translation are still under investigation and were discussed in detail in other review articles [146,147].

A tremendous scientific effort has focused on studying the toxicity of *C9orf72* G_4_C_2_^exp^ and dissecting the molecular mechanisms of C9 ALS/FTD. The overexpression of (G_4_C_2_)_66_ or (G_4_C_2_)_149_ using AAV approaches resulted in motor deficits, cortical neuron loss, gliosis and the accumulation of RNA foci and RAN proteins and TDP-43 pathology in mice [148,149]. Several bacterial artificial chromosome (BAC) transgenic mouse models of *C9orf72* G_4_C_2_^exp^ (C9-BAC) were generated to express the mutation using human, locus-specific promoters and to capture the possible involvement of flanking sequences surrounding the repeat [150,151,152]. While C9-BAC mouse models were reported to express molecular features of the disease including the accumulation of sense and antisense foci and RAN proteins, disease-related behavioral phenotypes varied significantly between models and within model. A direct comparison on RNA and RAN protein levels and accumulation patterns could provide an explanation for these differences. The genetic background and environmental factors could have also contributed to the manifestation of disease phenotypes in these BAC mouse models.

The overexpression of C9 dipeptides driven by the AUG start codon with G_4_C_2_^exp^ or alternative codons led to toxicity in cellular, zebrafish, *Drosophila* and mouse models [146,153]. Among five different C9 dipeptide expansion proteins, polyPR and polyGR were found to be the most toxic when overexpressed and mostly accumulated in the nucleus. C9 arginine-rich dipeptide proteins can dysregulate phase-to-phase separation, stress granule formation, RNA biogenesis, protein translation, ribosome function, and alternate extracellular matrix proteins [154,155,156,157]. C9orf72 polyGA was reported to adopt an amyloid-like structure that could act as a seeding template that recruits other misfolded proteins and promotes aggregation and co-aggregation with other proteins [158,159]. The proteasome function was impaired in polyGA-expressing cells and mice that led to the dysregulation of protein homeostasis. Additionally, polyGA was shown to sequester proteins important for mitochondrial function, contributing to neuronal degeneration [160]. Compared to polyPR, polyGR, and polyGA, polyGP and polyPA were found to be less toxic in disease models [146]. In addition to the mechanisms described above, other pathogenic mechanisms, including the impairment of the nucleocytoplasmic transport were reported in C9 ALS/FTD, and could contribute to the disease during the initiation or progression of the disease and were discussed in detail elsewhere [161]. While studying specific C9 RAN proteins provided insight into their potential contribution to the disease, it is important to consider their toxicity in the presence of other disease pathological signatures. The relative levels and interaction between C9 RNA and RAN repeat expansion products could modify toxicity and contribution in the disease of individual C9 mutant species.

**Therapeutic strategies for C9 ALS/FTD.** Given multiple lines of evidence supporting the gain of function of C9 expansion RNAs and the toxicity of C9 RAN dipeptide proteins, therapeutic approaches targeting expansion RNAs and C9 RAN proteins were developed and tested in disease models. The treatment of an antisense oligo (ASO) targeting sense rG_4_C_2_^exp^ transcripts reduced levels of RNA foci and RAN proteins, and mitigated behavioral deficits in patient cell lines and a C9-BAC mouse model [162,163]. From the report of the first rG_4_C_2_^exp^-targeting ASO, the antisense approach was further developed for more potent and specific therapeutic molecules. Stereopure oligonucleotides against rG_4_C_2_^exp^ developed by Wave Life Sciences were shown to protect C9 motor neurons from glutamate-induced toxicity [164], and the most promising molecule, WVE-004 selectively reduced repeat containing transcript and lowered C9 dipeptide levels [165]. rG_4_C_2_^exp^ targeting ASOs with a reduced number of phosphorothioate linkages in the backbone showed improved safety while maintaining their beneficial effects in increasing motor neuron survival in C9-BAC models [166]. The treatment of the most promising ASO5-2 led to decreased polyGP levels in a single C9 ALS patient individual. Two rG_4_C_2_^exp^ targeting ASOs BIIB078 (Ionis and Biogen) and WVE-004 (Wave Life Sciences) were studied for their efficacy in C9 patients. Unfortunately, the clinical trials of both therapeutic molecules were terminated in 2022 and 2023 due to a lack of efficacy in patients (https://investors.biogen.com/news-releases/news-release-details/biogen-and-ionis-announce-topline-phase-1-study-results, accessed on 5 May 2024, https://www.thepharmaletter.com/article/wave-life-sciences-ends-wve-004-program, accessed on 5 May 2024). The treatment of BIIB078 was found to reduce CSF polyGA and polyGP levels; however, increased Nfl levels were detected in patients. These results raise a question of whether targeting sense rG_4_C_2_^exp^ is sufficient to treat C9 ALS/FTD and what caused the increase in the neuronal damage marker in patients upon the BIIB078 ASO treatment. To explore an alternative ASO approach for C9 ALS, ASOs targeting antisense rG_2_C_4_^exp^ were also developed. A recent report by Rothstein and co-workers showed that rG_2_C_4_, but not rG_4_C_2_, targeting ASOs reduced TDP-43 pathology in C9 ALS patient induced pluripotent stem-cell (iPSC)derived neurons [167].

Both passive and active immunotherapies were developed to target polyGA for C9 ALS/FTD. An anti-GA antibody was reported to inhibit the cell-to-cell transmission of polyGA [168]. The treatment of human recombinant antibody targeting polyGA improved molecular, pathological and behavioral deficits in C9-BAC mice without changing the expansion transcript and RNA foci levels [169]. Additionally, the levels of other C9 dipeptide proteins were also reduced in C9-BAC mice treated with the anti-GA antibody. The immunization of GA peptides into mice led to the production of anti-GA antibodies and rescued motor phenotypes in mice overexpressing (G_2_C_4_)_149_ [170]. Due to the involvement of stress in favoring C9 RAN production and accumulation [171,172,173,174], the approaches targeting the integrated stress response were also explored to lower C9 RAN protein levels. For examples, the inhibition of double-stranded RNA-dependent protein kinase (PKR)activation and phosphorylation using dominant-negative PKR or metformin resulted in decreased polyGA and polyGP levels and improved neuronal survival and behaviors of C9 BAC mice without altering expansion RNA levels [174]. Future studies could focus on studying the safety and efficacy of C9 dipeptide protein-targeting approaches in C9 ALS/FTD patients.

Gene editing tools that excise the C9 G_4_C_2_^exp^ mutation could offer an optimal solution for C9 ALS/FTD. Several proof-of-concept examples of CRISPR/Cas9-mediated excision of C9 G_4_C_2_^exp^ were demonstrated *in vitro* using transfection methods [175,176,177]. Using gRNA pairs against the C9orf72 flanking sequences surrounding the repeats and CRISPR/Cas9 system and an AAV delivery method, the work by Meijboom and co-workers demonstrated a successful removal of C9 G_4_C_2_^exp^ in C9 mice and in patient-derived iPSC motor neurons [178]. The long-term effects in disease models, editing efficiency at single cell levels, editing specificity and safety would need to be addressed before moving this strategy forward to test in patients. Additional CRISPR/Cas9 molecules and other methods could be used to decrease off-target effects for C9 G_4_C_2_^exp^ targeting CRISPR/cas9 molecules [179].

## 6. TDP-43-Mediated Cryptic Splicing and ALS

The cytoplasmic inclusion and mislocalization of TAR DNA-binding protein 43 (TDP-43) encoded by *TARDBP* is a pathological hallmark of more than 95% of ALS and a subset of FTD and Alzheimer’s disease cases. Rare mutations in *TARDBP* are known to cause ALS (*TARDBP*-ALS) [180,181]. A higher frequency of TDP-43 inclusions was found in two familial *TARDBP*-ALS cases compared to sporadic ALS, suggesting a larger involvement of TDP-43 pathology in the disease in mutation carriers [181]. Other proteinopathies were reported to induce TDP-43 mislocalization [182,183]. The dysregulation of TDP-43 mRNA post-transcriptional processing, TDP-43 protein cleavage, phosphorylation and ubiquitination can also lead to TDP-43 aggregation and mislocalization [184].

TDP-43 is a heterogeneous nuclear ribonucleoprotein (hnRNP) that is ubiquitously expressed and mainly resides in the nucleus under normal conditions. TDP-43 plays important roles in regulating RNA splicing, transcription, and stabilization microRNA biogenesis. The structures, functions, and multiple aspects of its pathological contribution to the disease were discussed in detail in other review articles [184,185,186]. While TDP-43 is believed to regulate multiple important processes in cells and TDP-43 depletion in the nucleus being shown to cause mRNA misregulation of more than 1500 genes [187], the consequences of these misregulating events and their contribution to disease are just beginning to be understood.

**TDP-43 and *STMN2.*** Stathmin-2 or STMN2 pre-RNA was identified as an important target of TDP-43 by Melamed and co-workers [188]. In cells, TDP-43 binds to the first intron of STMN2 pre-mRNA to promote the exclusion of this intron to generate mRNA. The loss of TDP-43 in the nucleus was found to favor the inclusion of the cryptic exon 2a in a truncated mRNA, subsequently leading to reduced stathmin-2 protein levels. Decreased levels of full-length stathmin-2 mRNA were significantly associated with TDP-43 pathology and detected in TDP-43-associated sporadic ALS, C9 ALS, FTD and Alzheimer’s disease [188,189,190]. *STMN2* encodes stathmin-2, which regulates microtubule stabilization and is important for axon outgrowth, maintenance and regeneration [191,192]. Heterozygous and homozygous stathmin 2 knockout mice developed neuromuscular junction denervation, muscle atrophy and motor function deficits [193,194,195]. The transient loss of stathmin-2 by *Stmn2* shRNAs led to neuromuscular junction denervation in mice [195], suggesting that the lowering of the stathmin-2 protein at a given time during disease progression could result in substantial consequences. *STMN2* is also genetically linked with ALS. The non-coding CA repeat in *STMN2* was reported to be linked with an increased disease risk in sALS with repeat lengths of more than 24 showing the strongest association [196]. These results support the therapeutic strategies that restore the function of stathmin-2 in ALS and other stathmin-2 and TDP-43-related conditions (https://www.neurologylive.com/view/first-ever-study-of-stathmin-2-therapy-als-commences, accessed on 5 May 2024).

**TDP-43 and *UNC13A.*** Rare variants in *UNC13A* are associated with an increasing risk of ALS and FTD, and have been linked with TDP-43 pathology [197,198,199,200]. The *UNC13A* rs12608932 risk genotype was shown to link reduced survival in ALS patients and more affected upper motor neurons and other distinct clinical features in ALS patients [201,202,203,204]. The UNC13A protein plays a critical role in regulating the formation of synaptic vesicles and release of neurotransmitters [205]. Two back-to-back papers published in 2022 by Brown et al. and Ma et al. showed that TDP-43 is a regulator of the cryptic splicing of *UNC13A.* TDP-43 depletion in the nucleus resulted in increased levels of the *UNC13A* cryptic exon between exons 20 and 21 and, subsequently, reduced UNC13A protein levels [206,207]. ALS risk variants in *UNC13A* were shown to favor *UNC13A* cryptic exon inclusion. *UNC13A* cryptic exon containing RNA was detected in autopsy tissue samples from patients affected by ALS, FTD and motor neuron disease with the presence of TDP-43 pathology. In addition to TDP-43, other RNA-binding proteins including hnRNP L, hnRNP A1 and hnRNP A2B1, were reported to repress the cryptic exon inclusion of UNC13A mRNA [208]. Future work could focus on investigating the contribution of the loss of function of UNC13A and if *UNC13A* cryptic exon containing RNA and its mutant protein contribute to disease pathogenesis and the interplay between multiple RNA-binding proteins in regulating *UNC13A* splicing and cryptic exon inclusion.

## 7. Conclusions

It has been more than 150 years since ALS was first described and more than 30 years since the first ALS gene was identified. We have gained significant understanding of ALS genetics and molecule mechanisms in the past several decades. The findings from recent studies led to the first gene therapy for ALS to treat patients carrying *SOD1* mutations. Multiple therapeutic targets and intervention strategies for both familial and sporadic forms of ALS were also identified, bringing hope for the development of future treatments (Table 2). At the same time, many questions remain to address, including the efficacy and long-term effects of *SOD1* ASO, if there could be a treatment for C9 ALS/FTD, FUS ALS and if restoring stathmin-2 or UNC13A function using gene therapy or antisense oligos targeting unwanted cryptic slicing events or rescuing the function of TDP-43 in the nucleus or targeting ATXN2 could be beneficial for ALS patients. The genetics of sALS are still far from being fully understood; advances in sequencing technologies, data-processing tools and learning lessons from the known forms of ALS would help to accelerate the identification of novel risk ALS genes and mutations. In parallel, multiple omics big databases from patients and control cases are expected to continue to significantly contribute to the findings of novel pathogenic pathways and targets. With tremendous scientific efforts, perhaps effective treatments for individual ALS patients could be available in the near future.

## Figures and Tables

**Table 1 cells-13-00888-t001:** FDA approved drugs for ALS (order by year of approval).

Treatment	Drug Molecules	Mechanisms of Action	Year of FDA Approval
Rilutek	Riluzole (oral tablet)	Inhibiting the release of glutamate	1995
Nuedexta	Quinidine sulfate and dextromethorphan hydrobromide	Regulating neurotransmission, targeting bulbar functions	2010
Radicava	Edaravone	Removing reactive oxygen species and reducing oxidative stress	2017
Tiglutik	Riluzole (thickened liquid form)	Inhibiting the release of glutamate	2018
Exservan	Riluzole, (oral film)	Inhibiting the release of glutamate	2019
RELYVRIO	Sodium phenylbutyrate and tauroursodeoxycholic acid	Targeting reactive oxygen species and oxidative stress	2022
QALSODY^®^	Tofersen	Targeting SOD1 mRNA, reducing SOD1 protein levels	2023

**Table 2 cells-13-00888-t002:** Summary of ALS mutations discussed in this review article and related FDA approved drugs and potential therapeutic strategies.

Mutations/Genes	Year of Identification	Contributing to Familial or Sporadic ALS	Molecular Mechanisms	Therapeutic Strategies
*SOD1* mutations	First set of mutations identified in 1993 Increasing number of mutations identified over the years	~10–20% fALS ~2% sALS	1. Gain of function of mutant SOD1 protein. 2. Loss of wild-type SOD1 could contribute to onset, severity, and progression of disease	Antisense oligos targeting SOD1 mRNA
*FUS* mutations	First set of mutations reported in 2009 Increasing number of mutations identified over the years	~4–6% fALS <2% sALS	1. Gain of function of cytoplasmic FUS inclusions 2. Loss of function of nuclear FUS protein	Antisense oligos targeting FUS mRNA Molecules targeting post-translational modification for altering the localization of FUS protein
*ATXN2* CAG repeat with intermediate lengths	First reported in 2010 ALS risk mutation	Both familial and sporadic ALS	Increasing TDP43-mediated toxicity and FUS-related pathology	Lowering ataxin-2 protein levels using antisense oligo approach or CRISPR/Cas13 system
*C9orf72* G_4_C_2_ repeat expansion	Risk locus identified in 2006 G_4_C_2_ repeat expansion identified in 2011	10–50% of fALS 5–7% of sALS	1. RNA gain of function sense rG_4_C_2_ and antisense rC_4_G_2_ expansion RNAs 2. Toxicity of C9 RAN dipeptide proteins 3. C9orf72 haploinsufficiency could exacerbate other pathogenic events	1. Antisense oligos targeting sense rG_4_C_2_ and antisense rC_4_G_2_ expansion RNAs 2. Immunotherapy targeting C9 expansion proteins 3. Gene editing to excise C9 *C9orf72* G_4_C_2_^exp^
*STMN2* CA repeat polymorphism in *STMN2*	CA repeat polymorphism in *STMN2* identified in 2022	Reduced stathmin-2 levels observed in both fALS and sALS. CA repeat polymorphism reported to link with sALS	1. A downstream effect of TDP-43 depletion 2. Reduced levels of stathmin-2 resulting into motor function deficits in mouse models	1. Restoring stathmin 2 levels using gene therapy or other modulators 2. Targeting unwanted cryptic splicing events with antisense oligos 3. Restoring TDP-43 function in the nucleus
*UNC13A* mutations	Risk locus identified in 2009 Risk variants identified over the years	*UNC13A* mutations reported to link with sALS	1. A downstream effect of TDP-43 depletion 2. Reduced levels of UNC13A protein, which is important for neurotransmission	1. Restoring UNC13A levels using gene therapy or other modulators 2. Targeting unwanted cryptic splicing events with antisense oligos 3. Restoring TDP-43 function in the nucleus

## Data Availability

Not applicable.

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
