# Peer review of "Updates on Disease Mechanisms and Therapeutics for Amyotrophic Lateral Sclerosis"

_cells, 2024, doi:10.3390/cells13110888_

Round 1

Reviewer 1 Report

Comments and Suggestions for Authors

The authors focus predominantly on the molecular mechanisms and current therapeutic approaches in amyotrophic lateral sclerosis (ALS), with a particular emphasis on familial ALS. While the review article explores the ongoing pharmacological research and the underlying pathogenic mechanisms, it falls short of investigating or elucidating potential therapeutic drugs, even those not yet approved for clinical use. Moreover, the review leaves several pertinent questions unaddressed.

1.     To improve this review article, the following suggestions may be considered: Include citations within your text (e.g., lines 26-33 and elsewhere).

2.     Incorporate a table summarizing current therapeutic drugs that target the functions of microglia, astrocytes, or neurons.

3.     Given that C9orf72-ALS represents the largest subset of ALS patients, the discussion and introduction to this topic are insufficient.

These adjustments aim to enhance the article's comprehensiveness and relevance to the current state of ALS research.

Comments on the Quality of English Language

Minor editing of English language required

Author Response

Reviewer 1.

Comments and Suggestions for Authors

The authors focus predominantly on the molecular mechanisms and current therapeutic approaches in amyotrophic lateral sclerosis (ALS), with a particular emphasis on familial ALS. While the review article explores the ongoing pharmacological research and the underlying pathogenic mechanisms, it falls short of investigating or elucidating potential therapeutic drugs, even those not yet approved for clinical use. Moreover, the review leaves several pertinent questions unaddressed.

  1. To improve this review article, the following suggestions may be considered: Include citations within your text (e.g., lines 26-33 and elsewhere).

Response: The author thanks the reviewer for this helpful comment. The additional citations were included in the revised manuscript, including for the lines 26-33 in the previous version of the manuscript.

  1. Incorporate a table summarizing current therapeutic drugs that target the functions of microglia, astrocytes, or neurons.

Response: The author thanks the reviewer for this helpful comment. The author agreed that targeting functions of microglia, astrocytes or neurons is an interesting topic to discuss. Due to the focus on disease genetics and the development of therapeutics driven by genetics information, this review manuscript did not discuss this topic extensively. However, in the revised manuscript, the contribution of astrocytes and microglia to ALS were discussed in several ALS subtypes (e.g. FUS ALS).

  1. Given that C9orf72-ALS represents the largest subset of ALS patients, the discussion and introduction to this topic are insufficient.

Response: The author thanks the reviewer for this comment. The revised manuscript included additional information on C9ALS/FTD and cited other review articles that would provide comprehensive summaries of C9ALS/FTD

These adjustments aim to enhance the article's comprehensiveness and relevance to the current state of ALS research.

Thank you for the very helpful comments.

Reviewer 2 Report

Comments and Suggestions for Authors

The study entitled “Updates on disease mechanisms and therapeutics for amyotrophic lateral sclerosis” discusses the latest updates in ALS research on disease mechanisms and therapeutics. Here are my comments:

The abstract should be better revised. In fact, it is difficult for the general reader to understand the aims of the manuscript in this format. In the abstract section, the author should write in addition to the background of the research, what motivated the researcher to do this review and why this study is important.

In the second part of the Introduction to ALS section, the treatments available for ALS should be better presented, possibly with an easier to understand table.

The organization of sections presenting genes and therapeutic strategies should be consistent throughout the manuscript (SOD1, C9orf72  and TDP 43).

It is not clear to me why some genes (causal and/or risk factors) and associated current therapeutic strategies were chosen and not others. For example, why exclude genes such as FUS, ATXN2..? The author should clarify this.

The link: https://www.neurologylive.com/view/first-ever-study-of-stathmin-2-therapy-als-381 commences does not work.

Some of the numbered references are not associated with the appropriate paper (e.g. reference [5]).

In the Table 1, the author reports “….related potential therapeutic strategies”. There is some confusion on this point. Some therapies are already a reality, not a potential.

Comments on the Quality of English Language

There are many typos.  

Author Response

Reviewer 2.

Comments and Suggestions for Authors

The study entitled “Updates on disease mechanisms and therapeutics for amyotrophic lateral sclerosis” discusses the latest updates in ALS research on disease mechanisms and therapeutics. Here are my comments:

  1. The abstract should be better revised. In fact, it is difficult for the general reader to understand the aims of the manuscript in this format. In the abstract section, the author should write in addition to the background of the research, what motivated the researcher to do this review and why this study is important.

Response: The author thanks the reviewer for this comment. The goal of this review was included in the abstract in the revised manuscript. “The goal of this review article is to provide a comprehensive summary of the recently reported ALS mechanisms and related therapeutic strategies to the scientific audience”.

  1. In the second part of the Introduction to ALS section, the treatments available for ALS should be better presented, possibly with an easier to understand table.

Response: The author thanks the reviewer for this comment. To address this point, a new table 1 summarizing FDA approved drugs for ALS was included in the revision.

  1. The organization of sections presenting genes and therapeutic strategies should be consistent throughout the manuscript (SOD1, C9orf72  and TDP 43).

Response: The organization of sections of genes and therapeutic strategies were revised to be consistent in the revised manuscript.

  1. It is not clear to me why some genes (causal and/or risk factors) and associated current therapeutic strategies were chosen and not others. For example, why exclude genes such as FUS, ATXN2..? The author should clarify this.

Response: The author thanks the reviewer for this helpful comment. Discussion on FUS ALS and ATXN2 in ALS were included in the revised manuscript.

  1. The link: https://www.neurologylive.com/view/first-ever-study-of-stathmin-2-therapy-als-381-commences does not work.

Response: The link was updated in the revision. The correct link is https://www.neurologylive.com/view/first-ever-study-of-stathmin-2-therapy-als-commences

  1. Some of the numbered references are not associated with the appropriate paper (e.g. reference [5]).

Response: The author thanks the reviewer for this helpful comment. The reference was updated in the revised version to refer to the appropriate paper. For example, the references for for the sentence “Approximately 50% ALS develop frontotemporal dementia (FTD)-related features that are neuropathologically characterized by neuronal loss in frontal and temporal lobes and changes in behavior and judgement ability” are the following articles: https://pubmed.ncbi.nlm.nih.gov/28054827/ and https://pubmed.ncbi.nlm.nih.gov/28596248/.

  1. In the Table 1, the author reports “….related potential therapeutic strategies”. There is some confusion on this point. Some therapies are already a reality, not a potential.

Response: The author thanks the reviewer for this comment. The title of the table was edited as below.  “Summary of ALS mutations discussed in this review article and related FDA approved drugs and potential therapeutic strategies.”

Reviewer 3 Report

Comments and Suggestions for Authors

Major concerns:

Introduction: Regarding the non-motor symptoms of ALS, in addition to cognitive and behavioral disorders, also consider autonomic disorders (DOI: 10.1007/s00415-023-11832-w) and sensory/painful disorders (DOI: 10.1093/brain/awad426) which they can have a great impact on prognosis, survival and quality of life.

SOD1: in this paragraph I would also like to highlight the recent approval by the EMA for Tofersen, briefly describe the real experience of the German multicenter study (doi: 10.1016/j.eclinm.2024.102495). The author did not mention the ATLAS study nor the interesting study with adeno-associated virus encoding a microRNA targeting SOD1 (DOI: 10.1056/NEJMoa2005056). Please discuss all these points.

FUS gene: I would suggest including a paragraph dedicated to FUS gene therapy.

Minor concerns:

1.      The numbering of the references appears to be incorrect. Please, check.

2.      https://www.neurologylive.com/view/first-ever-study-of-stathmin-2-therapy-als-381 commences).--> the link does not work. Please check.

3.       The following sentence is confusing: “Furthermore, several 37 cases of spinal-onset ALS have been reported to have developed classic symptoms of myopic-38 ALS characterized by primary skeletal muscle injury.” I would suggest removing it.

Author Response

Reviewer 3.

Comments and Suggestions for Authors

Major concerns:

  1. Introduction: Regarding the non-motor symptoms of ALS, in addition to cognitive and behavioral disorders, also consider autonomic disorders (DOI: 10.1007/s00415-023-11832-w) and sensory/painful disorders (DOI: 10.1093/brain/awad426) which they can have a great impact on prognosis, survival and quality of life.

Response: The author thanks the reviewer for this helpful comment. Autonomic disorders and sensory/painful disorders were discussed in the revised manuscript.

  1. SOD1: in this paragraph I would also like to highlight the recent approval by the EMA for Tofersen, briefly describe the real experience of the German multicenter study (doi: 10.1016/j.eclinm.2024.102495). The author did not mention the ATLAS study nor the interesting study with adeno-associated virus encoding a microRNA targeting SOD1 (DOI: 10.1056/NEJMoa2005056). Please discuss all these points.

Response: The author thanks the reviewer for this helpful comment. These points were discussed in the revised manuscript in the SOD1 section.

  1. FUS gene: I would suggest including a paragraph dedicated to FUS gene therapy.

Response: The author thanks the reviewer for this helpful comment. Discussion on FUS ALS and related gene therapy were included in the revised manuscript.

Minor concerns:

  1. The numbering of the references appears to be incorrect. Please, check.

Response: The numbering of references was checked. Thank you for the comment!

  1. https://www.neurologylive.com/view/first-ever-study-of-stathmin-2-therapy-als-381 commences).--> the link does not work. Please check.

Response: The link was updated in the revision. The correct link is https://www.neurologylive.com/view/first-ever-study-of-stathmin-2-therapy-als-commences.

  1. The following sentence is confusing: “Furthermore, several 37 cases of spinal-onset ALS have been reported to have developed classic symptoms of myopic-38 ALS characterized by primary skeletal muscle injury.” I would suggest removing it.

Response: The author thanks the reviewer for the helpful comment. The sentence was removed in the revised manuscript.

Round 2

Reviewer 1 Report

Comments and Suggestions for Authors

No comments for authors.

Reviewer 3 Report

Comments and Suggestions for Authors

The author did a good job in reviewing the article, I have no further comments.